# Time as a significant factor in the release of potassium from lithium heparin plasma and serum

Tom Reuter[1,2]*, Michael Müller[2,3], Felix Stelter[2,4], Jürgen Durner[2,5,6☯], Jan Kramer[1,2,7☯]

**1** LADR Laboratory Group Dr. Kramer & Kollegen, Geesthacht, Germany, **2** Accredited Laboratories in Medicine (ALM) e.V., Quality Management Working Group, Berlin, Germany, **3** Medical Care Centre Labor 28, Berlin, Germany, **4** Laboratory Augsburg MVZ GmbH, Augsburg, Germany, **5** Department of Conservative Dentistry and Periodontology, University Hospital, LMU Munich Ludwig-Maximilians-University of Munich, Munich, Germany, **6** Laboratory Becker, München, Germany, **7** Division of Nephrology and Transplantation Unit, Department of Internal Medicine I, University of Lübeck, Lübeck, Germany

☯ These authors contributed equally to this work.
* t.reuter@LADR.de

**Data Availability Statement:** All relevant data are within the manuscript and its Supporting Information files.

**Funding:** The author(s) received no specific funding for this work.

## Abstract

### Objectives

In most countries the majority of patients are in outpatient care. In difference to hospitalized patients, their blood samples often take hours after collection to centrifugation. The study investigates the release of potassium and the development of pseudohyperkalemia in lithium heparin (Li-Hep) and serum blood collection tubes over time.

### Methods

From 201 donors 4 serum and 4 Li-Hep blood collection tubes were taken each. After 0.5, 4, 6 and 8h whole blood was centrifuged, and potassium levels were determined. To simulate the preanalytic conditions, the samples with a storage time >0.5h were shaken on a standard shaker for 1h and stored at 4–8°C for the remaining time.

### Results

Over time, significant more potassium was released before centrifugation from the Li-Hep plasma than from serum (1.21 vs 0.94 mmol/L). After 6h, the two groups were no longer highly statistically significantly different (potassium mean: 5.01 mmol/L in serum group, 4.92 mmol/L in Li-Hep group). In the Li-Hep group 164 donors developed a pseudohyperkalemia after 8h, compared to 76 in the serum group.

### Significance

The decision as to which material is best suited should not only be based on which value comes closest to the physiological situation immediately after blood collection. The subsequent preanalytic circumstances must also be considered. Serum tubes appear to be at least as suitable for potassium determination as Li-Hep tubes. In terms of patient blood management, serum provides the possibility of performing a wider range of analyses in the outpatient setting.

**Competing interests:** The authors have declared that no competing interests exist.

# 1 Introduction

Potassium is one of the most frequently requested analyses in clinical chemistry [1]. The analytically preferred material is heparinised whole blood, as no potassium is released from platelets during coagulation, as is the case with serum [2]. This is why the potassium values in serum are higher compared to lithium heparin (Li-Hep) plasma. This fact is sufficiently taken into account by material-dependent reference ranges [3]. In order to obtain reliable information on potassium values, it is recommended that heparin blood be kept at 25°C until centrifugation, which should take place within minutes after sampling. As this is very challenging, it is further discussed that centrifugation within one hour while keeping the heparin blood sample at room temperature will not introduce major errors [4].

The aforementioned extension of the processing time from minutes to one hour is only feasible in a few situations in the basic care of outpatients, e.g. for blood collection centres where centrifugation is available. Usually, several hours can pass when blood is collected in doctors' practices and then transported. Due to practice procedures and the shortage of staff, samples are only centrifuged in the minority of cases within time after the blood collection. In addition, there is no temperature control at 25°C, as cooling is necessary for some other analytes to ensure their stability.

Delayed measurement and improper storage can lead to hemolysis, which can result in pseudohyperkalemia and was first described by Hartmann *et al.* [2]. Potassium is one of the most frequently occurring exchangeable cations in humans. Its intracellular quantity in erythrocytes is stated to be around 100 mmol/L [5], in other cells about 140 to 150 mmol/L, the extracellular quantity as 3.5 to 5 mmol/L [6]. This gradient is important for the cell membrane potential and is maintained by the $Na^+/K^+$-ATPase [7].

The high intracellular concentration on the one hand and the number of erythrocytes on the other explain why the potassium level is susceptible to hemolysis. Other corpuscular components of the blood can also lead to an increase in potassium levels. This has been described for thrombocytosis [8] and leucocytosis [9]. Further reasons that might cause an increase in potassium levels is drawing blood by holding it in the upper arm for too long and "pumping" with the fist [10] or underfilling of tubes [11]. As described, the potassium value can be falsified, usually increased, by these factors. There is therefore a fear of creating pseudohyperkalemia, i.e. the potassium that is actually normal is increased by these factors. Some regional guidelines recommended that serum should be replaced by Li-Hep plasma, but this is not tested for different pre-analytical situations.

The aim of our study is to investigate whether Li-Hep plasma produces less pseudohyperkalemia than serum under conditions commonly encountered in general practice in Germany and other countries. These include the inability to centrifuge, longer storage times and vibration during transport. Our null hypothesis is that uncentrifuged Li-Hep plasma remains the collection material of choice over time under this common pre-analytical routine conditions.

# 2 Materials and methods

## 2.1 Participants

Patients from routine medical care were referred to the outpatient clinic of the DIN EN ISO 15189 accredited medical laboratory LADR MVZ Dr Kramer & Kollegen in Geesthacht, Germany. A favourable ethics vote from the University of Lübeck was obtained before the start of the study (No. 2023–746). The authors confirmed that they have complied with the World Medical Association Declaration of Helsinki regarding ethical conduct of research involving human subjects.

**Table 1. Overview of the randomly selected study population from the routine care of patients ≥18 years.**

| Number of patients | 201 |
|---|---|
| Gender | female: 111, Male: 90 |
| age | 54,1 y (SD 15,4y; range 20–88 y) |
| creatinine | 0,86 mg/dl (SD 0,18 mg/dl; range 0,44–1,61 mg/dl) |
| platelets | 268 G/l (SD 67 G/l; range 102–517 G/l) |
| leukocytes | 7,0 G/l (SD 2,9 G/l; range 3,1–36,5 G/l) |
| erythocytes | 4,8 T/l (SD 0,5 T/l; range 2,9–5,9 T/l) |
| hemolysis | hemolysis 0 (0–20 mg/dl) = 135; hemolysis 1 (21–99 mg/dl) = 65; hemolysis 2 (100–199 mg/dl) = 1 |

The patients were asked during the time period from 12[th] January 2024 to 5[th] March 2024 at the age of ≥18 years whether they would participate in a quality assurance study. They were informed and gave their consent for the additional nine blood samples to be taken for analysis. The age and gender of the person analysed were recorded. As part of the anonymised evaluation, it was not possible during the further process of the study to assign the materials and laboratory results to individuals, neither for the participants nor at the evaluation level. The leucocyte, platelet and erythrocyte counts showed no unexpected abnormalities, likewise the analyt creatinine (**Table 1**). No further exclusion or inclusion criteria were applied, as the group studied was intended to reflect the typical situation in an outpatient medical practice.

## 2.2 Preanalytical workflow including sample preparation and storage

The order in which the study tubes were taken from the volunteers was randomized and the tubes were completely filled. The following tubes were used for blood sampling (Greiner Bio-One, Frickenhausen, Germany): Vacuette® Z Serum 2.5 ml Seperator Coagulation activator, Vacuette® LH Lithium Heparin Separator 4 ml and Vacuette® K3E 1 ml EDTA.

After blood collection, blood tubes were left upright for 30 min. For the first time point of analysis (30 min) centrifugation was performed immediately afterwards and samples were measured from one of the serum and one of the Li-Hep plasma samples. Specimens were centrifuged using the Rotixa 50 RS Type 4910 (Hettich, Tuttlingen, Germany) at 3.500 × g for 10 min at room temperature (20–22˚C) prior to routine laboratory analysis.

Longitudinal laboratory analyses were in the course performed from the other (until then non-centrifuged) blood collection tubes at different time points, using a separate tube for each time point. These sample collection tubes for longitudinal analysis were initially stored at 4–8˚C without centrifugation as whole blood for a defined period of 3, 5 and 7 h respectively (simulation of sample storage in outpatient practices) and then the transport conditions in the car were simulated in a standardized manner for 1 h on a standard laboratory shaker RM5 (Karl Hecht, Sondheim, Germany) in the refrigerator at low speed.

Immediately afterwards (at 4 h after blood collection), the blood count was determined in the EDTA blood tube and, after centrifugation, potassium, creatinine and the Hemolysis, Icterus, and Lipemia (HIL) check were determined in the serum tube, and potassium and the HIL check were determined in the Li-Hep tube. 6 and 8 h after blood collection potassium was determined again in a Li-Hep plasma and serum tube (after centrifugation in each case immediately before the laboratory analysis). The potassium values at time 30 min and in the further longitudinal time course (4, 6 and 8 h) were compared accordingly within and between the two matrices and also evaluated in relation to other laboratory parameters such as leucocyte, erythrocyte and platelet count.

## 2.3 Potassium measurement

Laboratory analysis was perfomed on cobas® c systems (Roche, Mannheim, Germany) with CE-certified tests. The ion-selective electrodes (ISE) module of the Roche/Hitachi cobas c systems was used for the quantitative determination of potassium in serum or Li-Hep plasma using ISE with ISE indirect Na.K-Cl for Gen.2 (Roche).

## 2.4 Measurement of influencing factors

The Serum Index Gen.2 test (Roche) is based on calculations of absorbance measurements of diluted samples at different bichromatic wavelength pairs to semi-quantitatively detect hemolysis, icterus and lipemia (HIL check) in serum and plasma samples. The absorbance values for hemolysis are measured at 570 nm (primary wavelength) and 600 nm (secondary wavelength). The device calculates the index values from these absorbance values. The displayed and printed hemolysis index values do not have a unit. If the scale factors for conventional units are used, the displayed and printed values for hemolysis correspond to an approximate hemoglobin concentration in [mg/dL]. For determination of creatinine the Creatinine Jaffé Gen.2 (Roche) test application on cobas c systems was used. To produce the automated blood count with leucocytes, erythrocytes, erythrocyte indices and platelets, the dilution (Cellpack™) and staining (Fluorocell™) solutions recommended by the manufacturer were used on an XN series device system (Sysmex, Hamburg, Germany).

## 2.5 Data extraction and statistical analysis

The laboratory results were transferred online from the analyzer to the laboratory information system (LIS) molis (CompuGroup Medical, Koblenz, Germany) and documented there in anonymised form under a study order number. This internal barcode number was unrelated to the patient and did not allow retrospective allocation. The number was also used to identify the sample material by means of a barcode on the blood tubes, so that a clear assignment to the order was reliably possible in the routine. The results by parameter as well as the age and gender, which were also documented in the LIS, were digitally transferred to the Deltamaster program (Bissantz, Nürnberg, Germany) via the LIS data warehouse module and analysed in cubes before being exported to Excel (Microsoft, Redmond, Washington, US) for the final evaluation and graphical presentation of the data. For statistical analysis the Student's *t*-test was used. It is an established and recommend statistical technique used in the testing of hypothesis for comparison of the mean of two groups or whether two groups differ from each other [12]. The α-level for significance was 0.05 ($p < 0.05$)(raw data and statistical analysis can be found as file S1 Data).

# 3 Results

## 3.1 Time course of the potassium values

The absolute potassium values for each material at each time point of measurement after centrifugation immediately prior to analysis are shown in **Fig 1**. The first measurement of potassium concentration after 30 min showed higher potassium levels in serum than in Li-Hep plasma. The mean potassium concentration measured in Li-Hep plasma was 4.06 mmol/L (standard deviation (SD) 0.33, range from 3.1 to 5.4 mmol/L), which was highly significantly lower (p<0.001) than in serum with a concentration of 4.40 mmol/L (SD 0.35, range from 3.4 to 5.9 mmol/L).After 4 h, over 50% of the potassium values from Li-Hep plasma are out of the reference range.

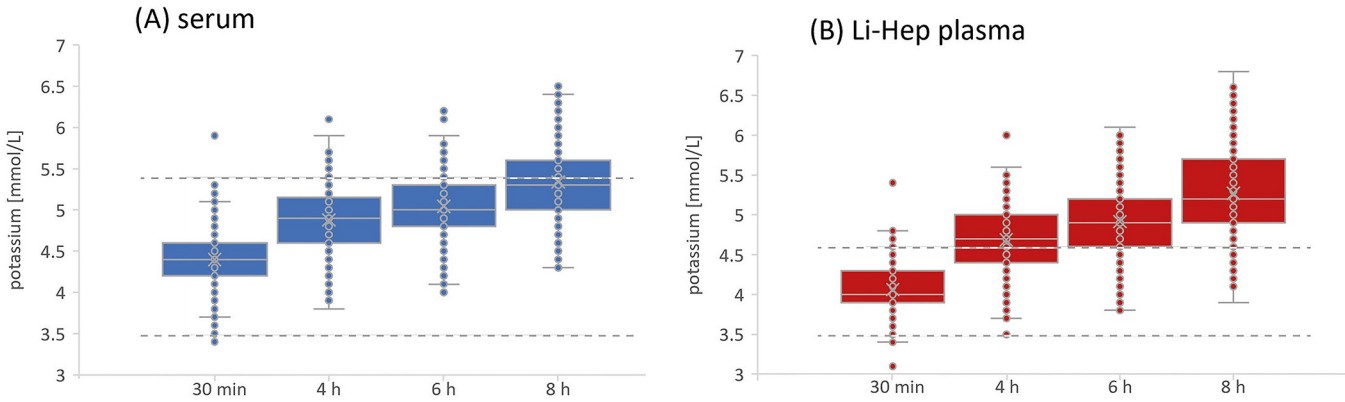

**Fig 1. Time course of the potassium values.** Serum (A, blue color boxplot) and Li-Hep plasma (B, red color boxplot) potassium values [mmol/l] are measured after 30 min, 4, 6 and 8 h. Whole blood was immediately centrifuged prior to analysis; dashed lines show the different reference intervals of both materials. Statistical data is given in the text (at section 3.1).

In time course, the absolute potassium values rised up in Li-Hep plasma and were no longer highly significantly different between the two materials after 6 h (p = 0.0024; <0.01) than at the time points after 30 min (p<0.001) and after 4 h (p<0.001). After 8 h, the potassium values were then equivalent in both materials (p = 0.084; not significant). The concentration measured in serum was 5.34 mmol/L (SD 0.46, range from 4.3 to 6.5 mmol/L) and 5.27 mmol/L (SD 0.54, range from 3.9 to 6.8 mmol/L) in Li-Hep plasma.

### 3.2 Higher increase of potassium concentration in Li-Hep plasma

An increase in potassium concentration was noted in both types of blood tubes as time progressed, shown as difference (delta) to the first measurement at 30 min in **Fig 2**. The change in mean potassium values was increasing in both materials from 4 h (serum: 0.48 mmol/L SD 0.18; Li-Hep plasma: 0.62 mmol/L SD 0.25) over 6 h (serum: 0.64 mmol/L SD 0.20; Li-Hep plasma: 0.86 mmol/L SD 0.29) up to 8 h (serum: 0.94 mmol/L SD 0.26; Li-Hep plasma: 1.21 mmol/L SD 0.39).

### 3.3 Higher number of pseudohyperkalemia in Li-Hep plasma

After 8 h, more than twofold of the Li-Hep plasma samples showed a pseudohyperkalemia compared to the serum samples (**Table 2**). This is due to a higher increase in Li-Hep plasma potassium concentration over time. Therefore, in the Li-Hep group 164 doners developed a pseudohyperkalemia after 8 h, compared to 76 in the serum group. The reference ranges are those for routine laboratory samples, in the case of potassium in serum 3.5–5.4 mmol/L and plasma 3.5–4.6 mmol/L.

The term 'pseudo' is used when there is a change in the reference range compared to the first time point measured after 30 min, e.g. if the potassium value was normal at 30 min and is outside the upper reference range after 8 h, the value is referred to pseudohyperkalemia for our data. No patient retained the hypokalemia.

### 3.4 No different impact of influencing factors on the potassium concentration in both materials

The number of platelets and the grade of hemolysis in our outpatient cohort had no significant impact on the potassium concentration after 30 min in serum and Li-Hep plasma (**Fig 3**). At

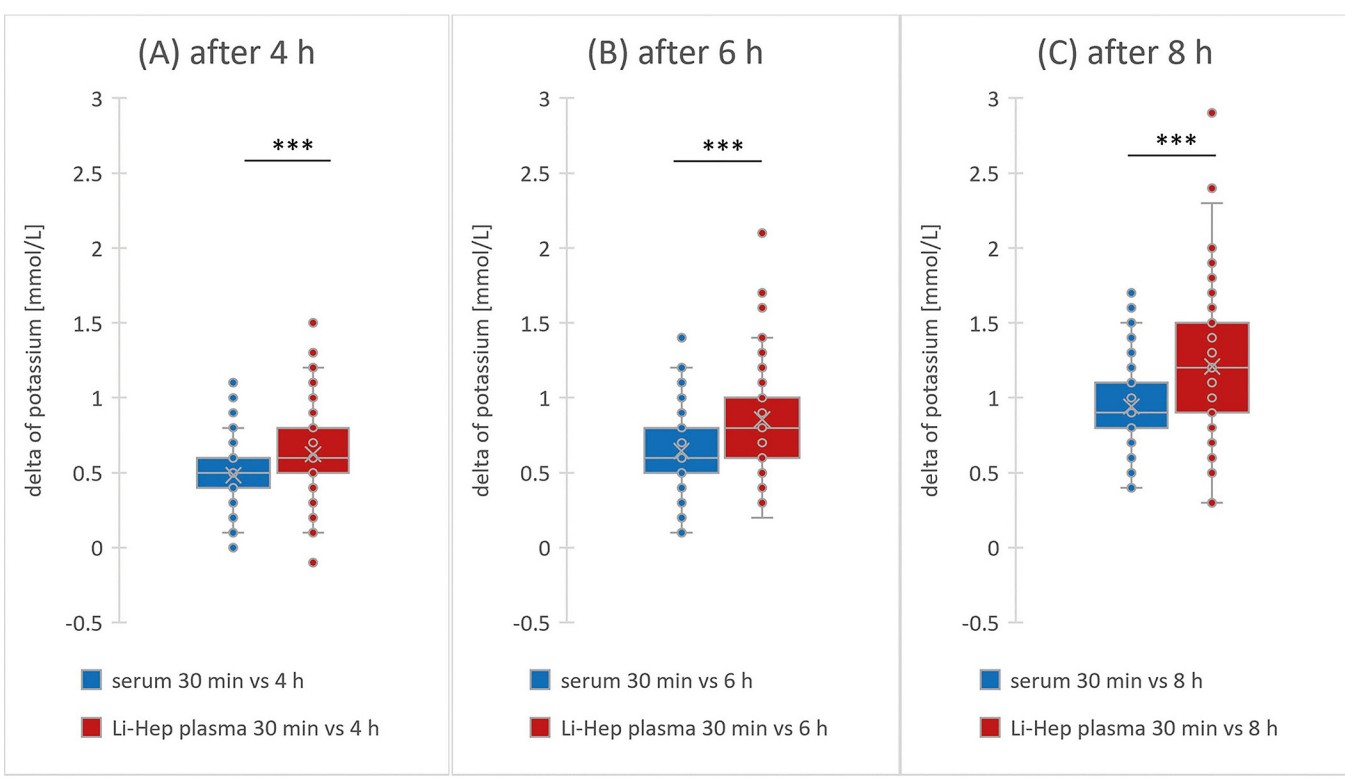

**Fig 2. Increase of potassium concentration.** Change in serum (blue color boxplot) and Li-Hep plasma (red color boxplot) potassium value after 4 h (A), 6 h (B) and 8 h (C) as difference (delta) to the potassium value after 30 min. Centrifugation was performed directly before measurement at the different time points. Significant difference: *** = p<0.001.

all timepoints and independent of number of platelets and grade of hemolysis, the change in potassium concentration was in Li-Hep plasma higher compared to serum (**Fig 4**).

The creatinine values of all patients are in the reference range. For these reasons, no patient had to be excluded. Overall, the change in the potassium concentration at various time points was independent of creatinine level and platelet-, leucocyte- and erythrocyte counts under the conditions tested in our study (**S1 Fig**).

## 4 Discussion

The influence of platelets, leucocytes and erythrocytes on the result of potassium measurement is well described [13]. It is known that the value of potassium in serum is higher than in Li-Hep plasma, which is why it is necessary to adapt the reference range to the different materials [3, 14, 15]. The potassium originates mainly from the platelets, which release it as part of the coagulation process. This is conform with the difference in potassium concentration between

**Table 2. Number of hypokalemia, normokalemia und hyperkalemia in serum and Li-Hep plasma after 30 min and after 8 h.**

|  | hypokalemia | normokalemia | hyperkalemia |
|---|---|---|---|
| serum after 30 min | 1/201 (0.5%) | 199/201 | 1/201 (0.5%) |
| Li-Hep plasma after 30 min | 5/201 (2.5%) | 187/201 | 9/201 (4.5%) |
| serum after 8 h | None (0%) | 124/201 | 77/201 (38.3%) |
| Li-Hep plasma after 8 h | None (0%) | 28/201 | 173/201 (86.1%) |

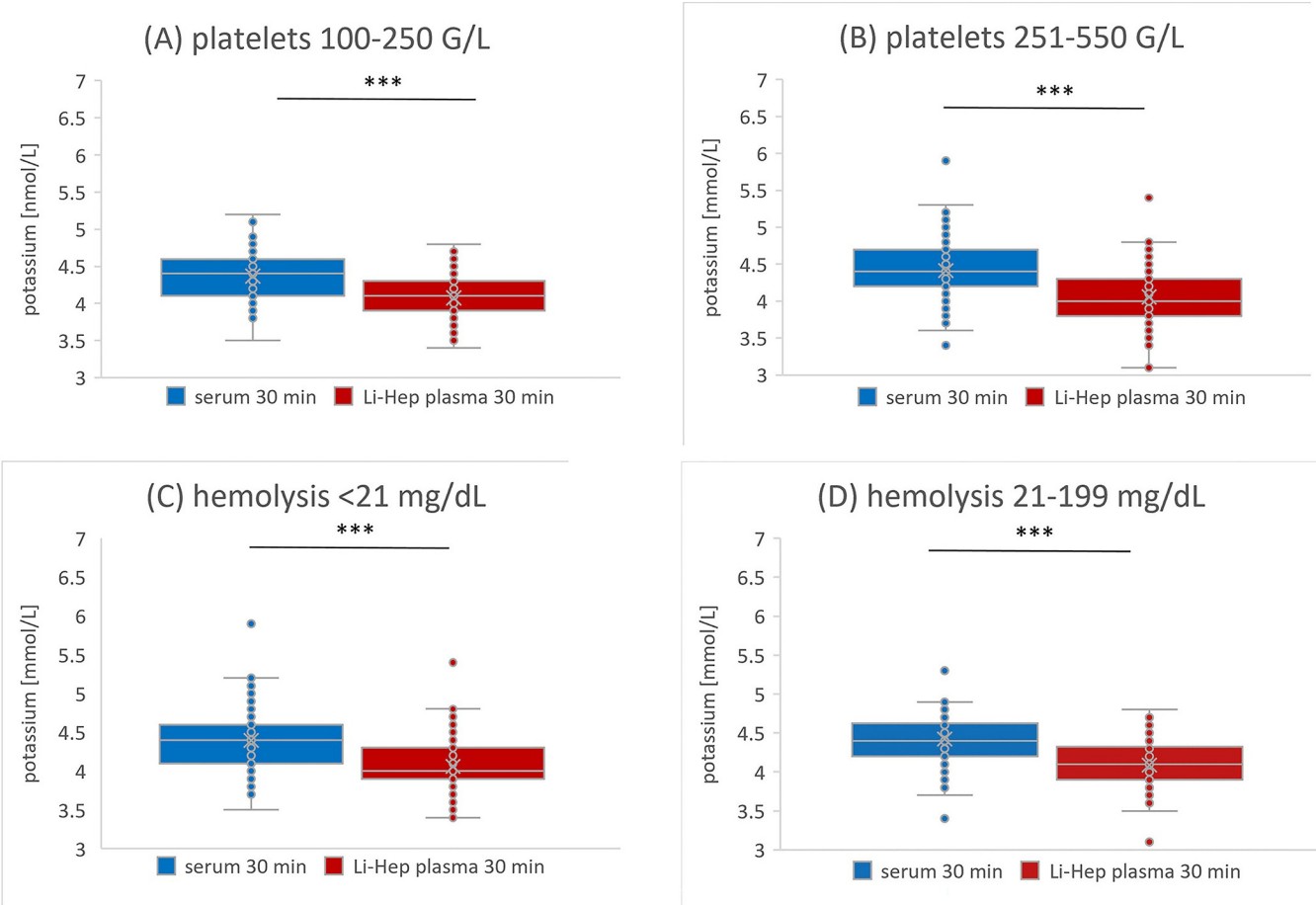

**Fig 3. Impact of influencing factors on baseline potassium concentration.** Serum (blue color boxplot) and Li-Hep plasma (red color boxplot) potassium value after 30 min divided by number of platelets (A,B) and grade of hemolysis (C,D). Significant difference: *** = p<0.001. There was no statistical significance within the group of serum samples or within the group of plasma samples in the comparison between A vs B or C vs D.

serum and Li-Hep plasma in our study at 30 min after sample taking. But already at this time a difference in eight pseudohyperkalemia were seen, one elevated in serum and nine in Li-Hep plasma (Table 2).

Increased platelet counts can lead to pseudohyperkalemia. One study showed that this becomes relevant from a platelet count of > 450 $10^9$/L [16]. Another study sees the limit at > 500 $10^9$/L [17]. A commentary should therefore be made as part of the medical validation as it is done in most laboratories. In a small cohort (n = 15) of patients with thrombocytosis no significantly increase in potassium in the plasma was observed compared to serum compared with a normal platelet count cohort (n = 13) [16]. In our study, no participant has a thrombocytosis and therefore our values were not falsified in this aspect. A generalisation to leucocytosis, especially in leukaemia patients, is not possible, as pseudohyperkalemia has been reported from both materials, although it appears that serum provides the correct values [18, 19].

In addition to these individual factors, the pre-analytical handling and additives in the blood collection containers also influence the analytical results [20]. Lippi *et al.* investigated the influence of storage 90 min after removal on various analytes. Blood was drawn from volunteers for serum and Li-Hep-based analyses and the blood was centrifuged. The blood collection tubes were from two manufacturers. In addition to the influence of the additives in the

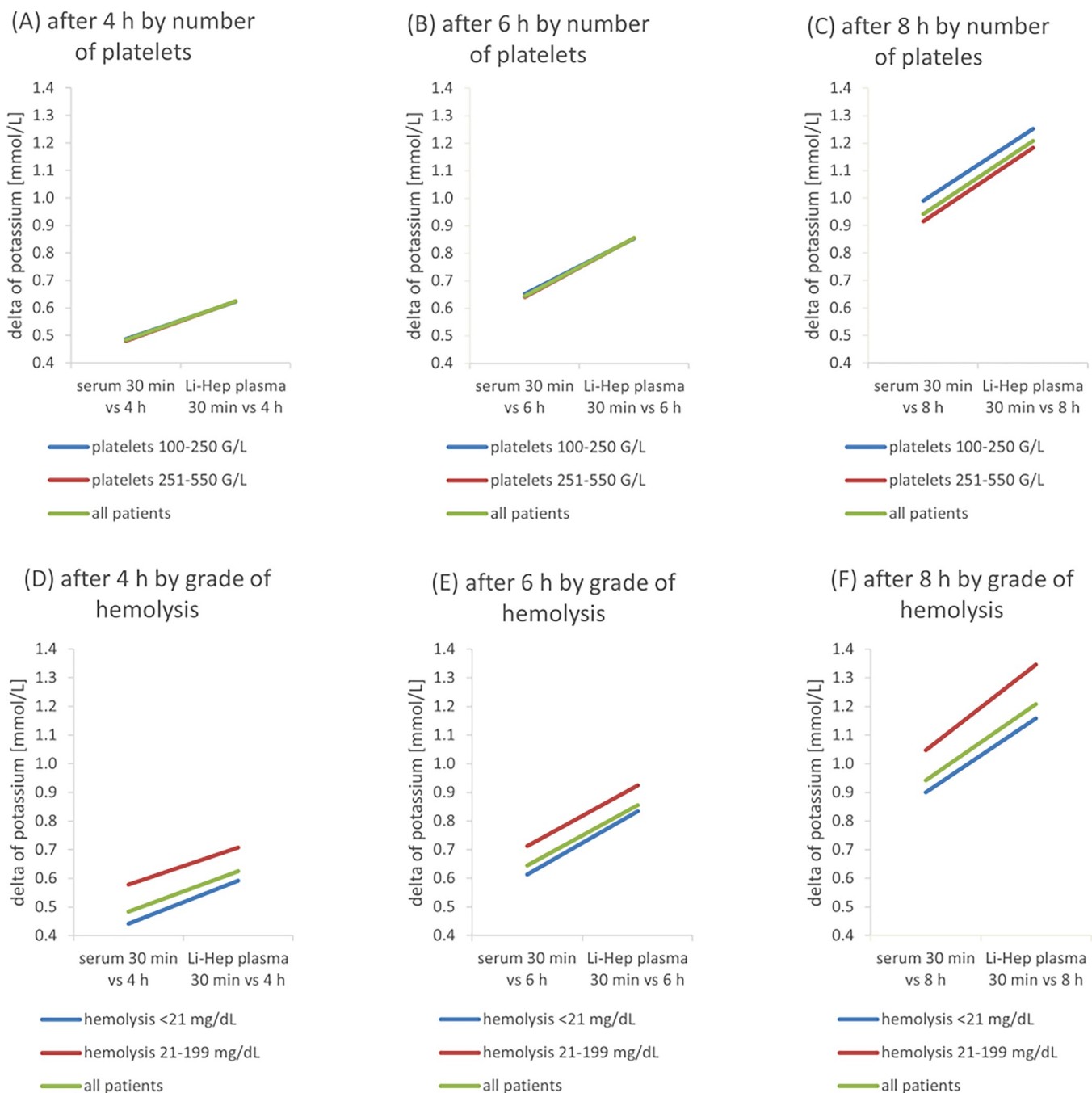

**Fig 4. Impact of influencing factors on potassium concentration at different timepoints.** Difference between the potassium value at 4 h (A, D), 6 h (B, E) and 8 h (C, F) and the potassium value after 30 min is shown for serum and Li-Hep respectively. The lines show the slope of the increase in mean between the materials as a function of the platelet count (A-C) and the grade of haemolysis (D-F).

blood collection tubes, the storage of the tubes was also analysed. In one group the blood collection tubes were stored vertically, in the other group they were rotated by 180° every 15 min [21]. A statistically significant increase in potassium levels was shown in the Li-Hep gel tubes, regardless of the manufacturer of the gel tubes, in this study, which is in good agreement with our findings. The increase of potassium levels was only shown for one manufacturer of serum gel tubes [21]. An increase in the hemolysis index was only observed in the case of the Li-Hep

gel tubes and not in the serum gel tubes, regardless of the manufacturer. The extent of hemolysis was determined by the HIL check in our cohort. The degree of hemolysis in both materials was mainly 0 or 1 out of 3 levels and thus did not influence the potassium values in a relevant way.

Another study investigated the stability of the analytes potassium, calcium and phosphorus in patients with essential thrombocytosis over a period of up to 8 h. Three different collection tubes were compared: serum with separator gel, Li-Hep plasma and Li-Hep plasma with separation gel [22]. In very good agreement with our results, the increase in potassium from Li-Hep plasma was also the strongest over the period up to 8 h. Besides potassium the other tested analytes only increased in Li-Hep plasma in a clinically relevant manner, according to published criteria from the European Federation of Clinical Chemistry and Laboratory Medicine Biological Variation Database [23].

The intra- to extracellular gradient is maintained by the $Na^+/K^+$-ATPase. The activity of this enzyme is temperature-dependent and it requires an appropriate nutritional supply. The increase in potassium, regardless of whether serum or Li-Hep plasma, is lower at 25°C compared to 4°C, which is due to the lower activity of the $Na^+/K^+$-ATPase at 4°C [24]. On the other hand, its normal activity leads to a decrease in the potassium concentration in serum and plasma at a long-lasting temperature of about 37°C and can cause spurious pseudohypokalemia. This effect is essentially observed in plasma samples in warm summer months [25, 26]. In particular in the outpatient sector the uncentrifuged samples are transported to the laboratory under refrigeration. To simulate transport refrigeration, the samples in our study were stored at 4–8°C, which leads to a limited function of the $Na^+/K^+$-ATPase. This is one reason why potassium increased more in Li-Hep plasma before centrifugation. However, cooling is important for the stability of other analytes and cannot be prevented.

The $Na^+/K^+$-ATPase obtains its energy from adenosine triphosphate (ATP), which is mainly produced by glycolysis and in the citric acid cycle [27]. Longer transport times lead to a drop in glucose and thus to a reduction in the activity of the enzyme [13]. The decrease of glucose over time is higher in Li-Hep plasma compared with serum [28]. The lack of energy for the $Na^+/K^+$-ATPase activity seems to be more important for Li-Hep plasma compared to serum. This is consistent with our study, in which well over 50% of Li-Hep plasma potassium values are outside the reference range within 4 h after blood taking.

In the Li-Hep plasma, there are almost only intact platelets and erythrocytes immediately after collection, but these can no longer transport the released potassium back into the cell over time due to the increasingly limited functionality of the $Na^+/K^+$-ATPase. For these reasons, the potassium concentrations in serum and Li-Hep plasma continue to converge over time. As a result pseudohyperkalemia or pseudonormokalemia can occur. Both effects were found in our study. The effect that normokalemia turned into pseudohyperkalemia was predominant, which was due to the initial situation of normokalemic subjects. The development of a pseudohyperkalemia is pronounced when using Li-Hep plasma in our outpatient cohort.

An important factor of the study is the simulation of non-stationary conditions, i.e. the sample material was processed in this study as in real care, including simulation of transport vibrations. In outpatient medical practices, the pre-analytical processes can only be standardized by the laboratory to a limited extent. For example, sample transport after blood collection during home visits or in nursing homes cannot be standardized in terms of temperature and time until centrifugation. The storage of samples in doctors' surgeries (room temperature versus refrigerator) cannot be influenced by the laboratory. This point must also be taken into account when recommending or selecting a specific sample material. The physiologically best values are not necessarily obtained under routine conditions, as our data show by the different generation of pseudohyperkalemia.

Pseudohyperkalemia can also occur when collection tubes are underfilled. Half-filled Li-Hep collection tubes caused 40% of the study participants to have pseudohyperkalemia [29]. Underfilled tubes play an important role in routine practice, especially in certain patient groups, like chemotherapy patients. The sole question of the sample material is extended by other (pre)analytical factors, some of which are the subject of current investigations [30–33].

The question arises as to whether the switch from serum to Li-Hep plasma has other disadvantages in the care of a large population. It was shown that free (f)T3 and fT4 are less stable in Li-Hep plasma. After 8 h, fT3 and fT4 should no longer be determined from Li-Hep plasma due to a shift in values, whereas the analyte remains stable in serum [11]. These analytes are often requested more than 8 h later from practising physicians, when they see that thyroid-stimulating hormone (TSH) is out of range e.g. during a guideline-based check of the thyroid gland function in adults [34]. In the inpatient area, the laboratory values are finished earlier (short transport and reporting time), so that subsequent requests are made earlier.

A similar time window for a possible subsequent determination like for fT3 from Li-Hep plasma is described for homocysteine, creatinine and other analytes like sex hormone-binding globulin (SHBG; stable in serum < 24 h and in Li-Hep plasma < 8 h). In the context of the step-by-step diagnostics of various diseases, additional requirements are indispensable. Here too, when switching to a different type of material (Li-Hep plasma vs. serum), it must be checked whether the recommended material can be used and if some analysis, like the electrophoresis from Li-Hep plasma, are impossible. The taking of both materials is not compatible with modern patient blood management.

A limitation of our study is that the centrifugation of the Li-Hep whole blood to plasma only took place immediately before the measurement. This was chosen because centrifugation is usually not possible in medical practices. In the literature, the use of a Li-Hep tube with separator gel also shows a greater increase in potassium compared to serum with separator gel [22]. The possibility of centrifugation of Li-Hep plasma with separation of the plasma into a separate tube immediately after blood collection would be the ideal solution, but in reality, this is not possible in the general population, as the workload in doctors' offices is too high and in home visits it is impossible. Depending on the situation in individual countries, it will not be possible to implement this approach, as the shortage of specialists will force us to work more efficiently, and this will be exacerbated by an increasing number of patients (demographic change). The focus on one analyte can be seen as a potential limitation. This is due to the fact that the aim was to specifically investigate whether there is a preferred sample material for potassium determination in the broad basic care provided by general practitioners and the associated conditions.

In addition to the analytical and physiological factors, pre-analytical factors must also be considered. Taking our results and relevant literature data into account, it can be stated for potassium determination that in settings where immediate centrifugation of the sample is possible, such as in hospitals and blood collection centres, Li-Hep plasma comes closest to the physiological potassium value. Our study showed, when samples are stored and transported uncentrifuged over a longer period of time, as may be the case in a wide patient care setting and supporting of practitioners, serum has been shown to produce less than half pseudohyperkalemia. According to these results, serum is the preferred material under these conditions, which is why the study data refute the null hypothesis.

## 5 Conclusions

In many cases it is not possible to process the samples promptly in the context of the real-life outpatient setting according to a lack of e.g. personal, time, equipment. In whole blood

samples before centrifugation the potassium release from Li-Hep plasma is significantly higher, and the superiority of Li-Hep plasma over serum cannot be demonstrated under the conditions of a routine outpatient care setting in this study. We found a more than twofold generation of pseudohyperkaemia in Li-Hep plasma compared with serum. On the laboratory side, a comment should therefore be made in two situations if there is a longer time between collection and centrifugation before measurement: In the case of normal potassium values at the lower reference range, pseudonormokalemia and in the case of high potassium values (over the reference range), pseudohyperkalemia cannot be ruled out.

The potassium value does not depend solely on the material taken, but also essentially on the accompanying circumstances. Li-Hep and serum both have their place in the current routine care of patients for the analysis of potassium and should be used selectively depending on the pre-analytical conditions. It is therefore not possible to make a clear recommendation for one sampling material. For the best possible care of the patient, there should therefore be a close dialogue between the laboratory and the doctor's office with the medical possibility to choose the optimal material for the corresponding situation.

## Supporting information

**S1 Fig. Impact of all analyzed influencing factors.** The difference between the potassium value after 8 h and 30 min is given as the delta for Li-Hep plasma (A, C, E) or serum (B, D, F). The potassium delta is set in relation to the influencing factors of the leukocyte (A, B), erythrocyte (C, D) and creatinine (E, F) measurement. $R^2$ is given as an indication of the quality of the linear regression. There is no linear correlation between X and Y in these data.
(TIFF)

**S1 Data. Raw data and statistical tests.** The file includes anonymized data of all measured patients including potassium concentration, influencing factors and age, as well as the statistical tests used.
(XLSX)

## Author Contributions

**Conceptualization:** Michael Müller, Felix Stelter, Jürgen Durner.

**Investigation:** Tom Reuter, Jan Kramer.

**Methodology:** Jürgen Durner, Jan Kramer.

**Project administration:** Jan Kramer.

**Resources:** Jan Kramer.

**Supervision:** Jürgen Durner, Jan Kramer.

**Writing – original draft:** Tom Reuter, Jürgen Durner, Jan Kramer.

**Writing – review & editing:** Michael Müller, Felix Stelter, Jürgen Durner, Jan Kramer.

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
