## [Decision Letter · Decision Letter 0]

20 Aug 2024

PONE-D-24-26828Time as a significant factor in the release of potassium from lithium heparin plasma and serumPLOS ONE

Dear Dr. Reuter,

Thank you for submitting your manuscript to PLOS ONE. After careful consideration, we feel that it has merit but does not fully meet PLOS ONE’s publication criteria as it currently stands. Therefore, we invite you to submit a revised version of the manuscript that addresses the points raised during the review process.

We look forward to receiving your revised manuscript.

Kind regards,

Abeer El Wakil, PhD

Academic Editor

PLOS ONE

Journal Requirements:

5. Please include your tables as part of your main manuscript and remove the individual files. Please note that supplementary tables (should remain/ be uploaded) as separate "supporting information" files

Reviewers' comments:

Reviewer's Responses to Questions

**Comments to the Author**

1. Is the manuscript technically sound, and do the data support the conclusions?

Reviewer #1: Partly

Reviewer #2: Partly

2. Has the statistical analysis been performed appropriately and rigorously? 

Reviewer #1: No

Reviewer #2: No

3. Have the authors made all data underlying the findings in their manuscript fully available?

Reviewer #1: Yes

Reviewer #2: Yes

4. Is the manuscript presented in an intelligible fashion and written in standard English?

Reviewer #1: No

Reviewer #2: No

5. Review Comments to the Author

**Reviewer #1:** While this topic is interesting, unfortunately, the article is not clearly written and many parts read more like a review than a research paper. The main issues are summarized below:

1.The introduction section of the article fails to reflect the purpose of the study and does not follow a clear flow. Some paragraphs are written in a review-like manner.

2. Similarly, the materials and methods section includes many details that are not helpful to the reader and lacks a clear methodological flow.

3.The results are not easily understandable.

4.The discussion section falls short of highlighting and explaining the main findings of the study and comparing them with the existing literature. It is written more like a review.

**Reviewer #2:** The article titled "Time as a Significant Factor in the Release of Potassium from Lithium Heparin Plasma and Serum" addresses a crucial aspect of clinical practice in human medicine. However, there are several areas that require improvement:

1. Introduction: The research objective is not clearly stated, making it difficult to understand the study's purpose.

2. Materials and Methods: This section is challenging to follow and should be restructured to offer a clearer, more comprehensive overview of the study's procedures. I recommend adding subheadings to enhance clarity and meet the high standards expected in scientific research. Additionally, the sample size and experimental procedures need to be clearly outlined.

3. Statistical Analysis: The statistical methods are not well-explained. Although the use of the Student’s t-test is mentioned, it’s unclear whether assumptions of normality and equal variances were tested. Furthermore, while the results discuss a correlation between changes in potassium concentration and renal function, as well as blood parameters, there is no explanation of the statistical tests used for these analyses. Additionally, there is no comparison of potassium concentrations at different centrifugation times (30 min, 4 h, 6 h, 8 h), which would have been valuable.

4. Results: Some figures are unclear. Specifically, in subsection 3.5 (lines 194-200), there is no explanation of the correlation coefficient (r value) or its significance.

6. PLOS authors have the option to publish the peer review history of their article (what does this mean?). If published, this will include your full peer review and any attached files.

Reviewer #1: No

Reviewer #2: No

---

## [Author Response · Author response to Decision Letter 0]

10 Oct 2024

Dear reviewer,

please refer to the attached rebuttal letter for a detailed response to the comments.

---

## [Decision Letter · Decision Letter 1]

28 Oct 2024

Time as a significant factor in the release of potassium from lithium heparin plasma and serum

PONE-D-24-26828R1

Dear Dr. Reuter,

We’re pleased to inform you that your manuscript has been judged scientifically suitable for publication and will be formally accepted for publication once it meets all outstanding technical requirements.

Kind regards,

Abeer El Wakil, PhD

Academic Editor

PLOS ONE

Additional Editor Comments (optional):

Reviewers' comments:

Reviewer's Responses to Questions

**Comments to the Author**

1. If the authors have adequately addressed your comments raised in a previous round of review and you feel that this manuscript is now acceptable for publication, you may indicate that here to bypass the “Comments to the Author” section, enter your conflict of interest statement in the “Confidential to Editor” section, and submit your "Accept" recommendation.

Reviewer #2: All comments have been addressed

2. Is the manuscript technically sound, and do the data support the conclusions?

Reviewer #2: Yes

3. Has the statistical analysis been performed appropriately and rigorously? 

Reviewer #2: Yes

4. Have the authors made all data underlying the findings in their manuscript fully available?

Reviewer #2: Yes

5. Is the manuscript presented in an intelligible fashion and written in standard English?

Reviewer #2: Yes

6. Review Comments to the Author

Reviewer #2: Authors have revised their manuscript and follow the reviewer suggestion. Therefore the manuscript much better and clear. It is now acceptable for publication

7. PLOS authors have the option to publish the peer review history of their article (what does this mean?). If published, this will include your full peer review and any attached files.

Reviewer #2: No

---

## [Editor Report · Acceptance letter]

12 Nov 2024

PONE-D-24-26828R1 

PLOS ONE

Dear Dr. Reuter, 

I'm pleased to inform you that your manuscript has been deemed suitable for publication in PLOS ONE. Congratulations! Your manuscript is now being handed over to our production team.

Kind regards, 

on behalf of

Professor Abeer El Wakil 

Academic Editor

PLOS ONE